# Modulation of the mTOR Pathway by Curcumin in the Heart of Septic Mice

**DOI:** 10.3390/pharmaceutics14112277

**Published:** 2022-10-24

**Authors:** Bruna A. C. Rattis, Henrique L. Piva, Andressa Duarte, Frederico G. F. L. R. Gomes, Janaína R. Lellis, Danilo F. Soave, Simone G. Ramos, Antonio C. Tedesco, Mara R. N. Celes

**Affiliations:** 1Department of Pathology, Faculty of Medicine of Ribeirão Preto, University of São Paulo, Ribeirão Preto 14040-900, São Paulo, Brazil; 2Department of Bioscience and Technology, Institute of Tropical Pathology and Public Health, Federal University of Goias, Goiânia 74605-050, Goias, Brazil; 3Department of Chemistry, Faculty of Philosophy, Science and Letters of Ribeirão Preto, University of São Paulo, Ribeirão Preto 14040-901, São Paulo, Brazil; 4Department of Morphofunctional, Faculty of Medicine of Goianesia, University of Rio Verde, Goianesia 76380-000, Goias, Brazil

**Keywords:** sepsis, mTOR, curcumin

## Abstract

mTOR is a signaling pathway involved in cell survival, cell stress response, and protein synthesis that may be a key point in sepsis-induced cardiac dysfunction. Curcumin has been reported in vitro as an mTOR inhibitor compound; however, there are no studies demonstrating this effect in experimental sepsis. Thus, this study aimed to evaluate the action of curcumin on the mTOR pathway in the heart of septic mice. Free curcumin (FC) and nanocurcumin (NC) were used, and samples were obtained at 24 and 120 h after sepsis. Histopathological and ultrastructural analysis showed that treatments with FC and NC reduced cardiac lesions caused by sepsis. Our main results demonstrated that curcumin reduced mTORC1 and Raptor mRNA at 24 and 120 h compared with the septic group; in contrast, mTORC2 mRNA increased at 24 h. Additionally, the total mTOR mRNA expression was reduced at 24 h compared with the septic group. Our results indicate that treatment with curcumin and nanocurcumin promoted a cardioprotective response that could be related to the modulation of the mTOR pathway.

## 1. Introduction

Sepsis-induced myocardial dysfunction (SIMD), a central component of Multiple Organ Dysfunction Syndrome (MODS) in sepsis, is associated with poor prognosis and higher mortality [1,2,3,4]. Several mechanisms are thought to be involved in this cardiac pathology, including tissue hypoxia, loss of structural proteins, impairment of contractile machinery, and dysregulation of molecular pathways essential for cardiomyocytes [5,6,7,8,9,10,11].

The mammalian target of rapamycin—mTOR pathway encompasses a variety of upstream signaling pathways responsible for cellular energy status, oxidative stress, amino acid availability, insulin, and growth factors, which play an essential role in cell growth, promotion of apoptosis, control of autophagy, and regulation of the actin cytoskeleton [12,13]. mTOR is a serine-threonine kinase, belonging to the phosphatidylinositol-3-kinase family (PIKKs) that forms the catalytic subunit of two distinct protein complexes (Figure 1) [14,15], which regulate anabolic processes, such as protein synthesis, lipids and nucleotides, and catabolic processes, such as autophagy and cell survival pathways [12,16].

The functions of the mTORC1 and mTORC2 complexes are distinct; mTORC1 integrates information about nutrient availability and environmental conditions and controls anabolic processes, such as protein, lipid, and nucleotide synthesis, and catabolic processes, such as autophagy [12,16], while mTORC2 regulates the cytoskeleton and cell survival pathways [12,16]. mTOR plays a central role in inflammatory processes and sepsis [17,18,19,20]. In the heart, mTOR regulates cardiomyocyte metabolism and cardiac remodeling, and is indispensable during embryonic cardiac development [21,22]. In experimental sepsis, pharmacological inhibition of mTOR was shown to be cardioprotective, which can be explained by the acceleration of autophagy [23].

Curcumin is a phytochemical with a wide variety of biological properties, including anti-inflammatory, antioxidant, antineoplastic, antiviral, antibacterial, and antifungal activities [24,25,26]. The effects of curcumin on the mTOR pathway were discovered in studies of its growth-inhibiting effects on neoplastic cells in rhabdomyosarcoma (Rh1 and Rh30) and rectal colon cancer (HCT116 and KM20), in which curcumin was shown to inhibit the mTORC1 complex [27,28]. To date, many studies have explored curcumin as an anti-inflammatory agent based on cytokine dosage and immune cell population in sepsis; however, there are no reports of its in vivo effect on the mTOR pathway in any pathology, especially in sepsis [29]. Thus, this work aims to fill this gap, being the first to evaluate the expression of mTOR components in an animal model. Therefore, this work aimed to understand whether in vivo curcumin, both in its free and nanocurcumin form, affects the mTOR pathway in the heart of septic mice.

Despite advances in our understanding of sepsis, there are few therapeutics for SIMD. Here, we investigated the effects of curcumin on the mTOR pathway in the hearts of mice with experimental sepsis induced by ligation and perforation of the cecum (CLP).

## 2. Materials and Methods

### 2.1. Experimental Animals

Male C57BL/6 mice, weighing 22–24 g, were kept at room temperature (°C) under a 12/12 h light-dark cycle. They were housed in the vivarium of the Department of Pathology of the Faculty of Medicine of Ribeirão Preto and received standard mouse chow and water ad libitum. The animal protocol was approved by the Animal Research Committee of the Faculty of Medicine of Ribeirão Preto, University of São Paulo, Brazil (Protocol n° 113/2019). Every effort has been made to minimize the suffering of animals. For the experiments, mice were arbitrarily allocated to six groups (Table 1).

Nanocurcumin was obtained and provided by Prof Dr. Antonio Claudio Tedesco from the Center for Nanotechnology and Tissue Engineering at the University of São Paulo (Appendix A). The present study followed the protocol to obtain nanocurcumin as published by BHAWANA et al. (2011), which demonstrated that freeze-dried nanocurcumin is stable at room temperature for six months without any decomposition or aggregation. Free curcumin (FC) (Sigma-Aldrich Co., St. Louis, MO, USA) and nanocurcumin (NC), both at a dose of 12.5 mg, were diluted in sterile 0.9% NaCl saline solution (100 uL volume/animal) and homogenized prior to administration, which occurred subcutaneously (s.b.) on the back of the animal shortly after CLP surgery or sham operation. Untreated septic mice and untreated control mice received an equivalent volume of saline. Survival rates were monitored every 12 h for five days after surgery, using eight animals per group.

### 2.2. Polymicrobial Sepsis (Cecal Ligation and Puncture—CLP Model)

For the induction of experimental sepsis, the protocol of induction of polymicrobial sepsis was followed from the model of ligation and perforation of the cecum [29,30]. The mice were rapidly anesthetized with 2.0–3.0% isoflurane and vaporized in medical oxygen (O_2_), through a face mask. The abdomen was shaved and a midline incision was made. The cecum was isolated and ligated with a 6–0 silk thread below the ileocecal valve without causing intestinal obstruction. The cecum was then punctured with an 18 G needle to induce severe septic stimulus (CLP). The intestinal contents were gently extruded through the puncture and the cecum was returned to its original position. The abdomen was then sutured. Sham-operated animals (controls) were submitted to the same procedures, except for cecal ligation and puncture. Immediately after surgery, each animal received a subcutaneous injection of 1 mL of saline and was placed in an incubator at 37 °C for 30 min. Postoperatively, all animals received treatment with centrally acting analgesic Tramadol (Tramal^®^) at a dose of 12.5 mg intramuscularly after suturing and 5 mg/kg diluted in water available in the drinkers of each cage throughout the experimental period of 120 h.

Mice were monitored daily for signs of illness, such as piloerection, stooped gait, lethargy, eye discharge, and diarrhea following the recently revised murine sepsis endpoint [31]. Mice that exhibited severe signs of distress (difficulty breathing, unresponsiveness to cage tapping, poor hygiene, severe eye discharge) were euthanized by injection of a mixture of ketamine (90–120 mg/kg) and xylazine (10 mg/kg), followed by cervical dislocation.

### 2.3. Euthanasia and Collection

After 24 and 120 h of sepsis induction, the animals in each group were anesthetized with 2.0–3.0% isoflurane and vaporized in medical oxygen (O_2_) through a face mask. The chest cavity was opened, exposing the still-beating heart. Aortic exsanguination was performed, then the hearts were quickly excised and washed with ice-cold saline (4 °C) of 0.9% NaCl, and dried on filter paper. Subsequently, the hearts were sectioned longitudinally into two halves. Heart samples were placed in 10% buffered formalin or frozen at −80 °C for Western Blotting (WB) or polymerase chain reaction (PCR). It is worth mentioning that a small fragment of 1.0–0.5 mm fixed in glutaraldehyde was removed. The other collected organs had fragments directed to the WB stored at −80 °C and fixed in buffered formalin 10 for conventional optical microscopy.

### 2.4. High-Resolution Microscopy

Heart samples were dehydrated in increasing concentrations of alcohol 70, 95 (3 exchanges of 15 min), and 100% (3 exchanges of 1 h each), passed through the preinfiltration solution (24 h), infiltration solution (24 h), embedded in resin, and placed on appropriate supports (Historesin^®^, Leica Instruments GmbH, Heidelberg, Germany). The material was incubated for 24 h in an oven at 60 °C to harden the resin. Sections measuring 2.5 μm were obtained on a Sorvall JB4-A microtome (DuPont Company, Wilmington, DE, USA), stretched in a water bath at room temperature, placed on a glass slide, and dried on platinum heated to a temperature of 55–60 °C for approximately 24 h. Soon after, the sections were stained with toluidine blue and evaluated under a Leica DMRS microscope (Leica Microsystems, Wetzlar, Germany).

### 2.5. Transmission Electronic Microscopy

The characterization of the ultrastructural alterations of the hearts of the different groups of animals submitted to sepsis was evaluated by transmission electron microscopy. The samples were fixed by immersion in 3% glutaric aldehyde in 0.1 M phosphate buffer, pH 7.3 at 4 °C, for fixation. After washing in 0.1 M phosphate buffer, the tissue was postfixed in 1% osmium tetroxide solution in 0.1 M phosphate buffer at 4 °C. The tissue was dehydrated in increasing concentrations of acetone and embedded in Araldite. Subsequently, ultrathin sections were made, obtained with a diamond knife, which was stained in uranyl acetate and lead citrate to be examined and photographed under a transmission electron microscope.

### 2.6. Western Blotting

Collected heart samples were immediately frozen at −80 °C until protein extraction. For protein extraction, 400 μL of extraction buffer (100 mM NaCl, 10 mM Tris-Cl, pH 7.6, 0.1% SDS, and 1 mM PMSF; Sigma-Aldrich) were added to the tissue, which was ground with a fabric (DRE-MEL^®^300, Marconi, Ribeirão Preto, Brazil). After this procedure, the material was centrifuged at 12,000 rpm for 20 min at 4 °C. The supernatant containing the protein solution was collected and aliquoted for protein dosage. Protein dosage was performed in a spectrophotometer at a wavelength of 595 nm by the Bradford method. Then, samples containing 40μg of proteins were applied on SDS-PAGE electrophoresis gel at 7 or 10% according to the molecular weight of the protein to be analyzed. After the gel run, the proteins were transferred to a PVDF membrane (Immobilon^®^-Psq, Millipore Corporation, Billerica, MA, USA). Complete protein transfer was confirmed by gel and membrane staining with Ponceau S stain (Sigma-Aldrich, St. Louis, MO, USA). The PVDF membrane carrying the proteins was placed in a blocking solution with 5% BSA (Sigma-Aldrich) overnight at 4 °C. Afterwards, the membrane was washed in PBS-T (phosphate buffered saline) with Tween-20 (pH 7.2–7.4) and incubated overnight at 4 °C with the primary antibody: anti-mTOR, anti-GAPDH (Cell Signaling) diluted in 1% BSA solution, and PBS-T buffer. After incubation, the membranes were washed in PBS-T and incubated for 45 min at room temperature with HRP-conjugated secondary antibodies. α-rabbit-IgG/HRP antibodies were used. Then, the membrane was washed again in PBS-T buffer and incubated with Immobilon Forte Western HRP substrate (Millipore) developing a solution according to the manufacturer’s specifications. After this process, the membrane was developed in the ChemiDoc XRS device (BioRad). The quantification of specific bands was performed using the public domain ImageJ Program (developed at the National Institutes of Health). The reported values refer to the optical density (OD) of the bands expressed in arbitrary units (AU).

### 2.7. RNA Isolation, Reverse Transcription, and RT-PCR

Total RNA was isolated from the heart (n = 6 animals/group) using the TRIzol reagent (Invitrogen, Carlsbad, CA, USA). Real-time quantitative polymerase chain reaction (qPCR) of cDNA with the SYBR II Green QPCR was performed with glyceraldehyde-3-phosphate dehydrogenase (GAPDH) as an internal control. The nucleotide sequences of the primers from previous studies were as follows: mTORC1, mTORC2, Raptor, and Rictor (Table 2). For the mTOR gene, TaqMan probes (TaqMan Universal PCR Master Mix, Applied Biosystems, CA, USA) were used. The reaction products were amplified using the GoTaq qPCR Master Mix kit (Promega, Madison, WI, USA) under the following conditions: polymerase activation (1 cycle) at 95 °C for 2 min, denaturation and annealing (40 cycles) at 95 °C for 15 s, and 65 °C for 1 min, dissociation (1 cycle) 60–95 °C. Quantification of mTOR, mTOCR1, mTORC2, Raptor, and Rictor gene expression was expressed in ΔΔCT values.

### 2.8. Statistical Analysis

The data obtained were analyzed using the GraphPad Prism 8 statistical program (Graph Pad Software In., San Diego, CA, USA). Student’s *t*-test was used to compare two normally distributed variables. Multiple comparisons were made by analysis of variance (ANOVA) followed by the Bonferroni post-hoc test. The survival rate was presented as the percentage of live animals. The significance level chosen was 5% and data are presented as mean + standard error of the mean (SEM).

## 3. Results

### 3.1. Characterization of Murine Sepsis Using the CLP Model

The mice in the control groups (SHAM, SH + FC, and SH + NC) had no differences in weight, temperature, or behavior. Weight and temperature also did not differ between the mice in the untreated septic group (CLP and curcumin-treated septic mice (CLP + FC and CLP + NC)). However, the mice in the untreated septic group (CLP) exhibited physiological and behavioral changes, including piloerection, prostration, little or no response to stimuli (sound and tactile), gasping for breath, semi-open eyes with ocular secretions, diarrhea, and progressive weight loss. Compared to the CLP group, the sepsis-model mice treated with free curcumin (CLP + FC) and nanocurcumin (CLP + NC) showed milder behavioral changes.

In the first period of evaluation (6 h), no physical changes were observed among the groups, and there were no deaths. At the end of the 120-h observation period (five days), the survival of the sham-operated mice (SHAM, SH + FC, and SH + NC) remained at 100%. In contrast, after five days, approximately 50% of the septic mice in the untreated (CLP) and nanocurcumin-treated (CLP + NC) groups survived, whereas approximately 80% of the septic mice treated with free curcumin (CLP + FC) survived.

### 3.2. Effects of Curcumin on Cardiac Morphology

#### 3.2.1. Histopathological Evaluation of the Heart

The control groups (SHAM, SH + FC, and SH + NC) showed no morphological changes in the heart (Figure 2). However, histopathological analysis of the myocardium of the mice in the CLP group at 24 and 120 h showed diffuse foci of myocytolysis, edema, and focal contracture bands. The mice in the CLP + FC group had cytoplasmic vacuoles at 24 h and foci of myocytolysis and cytoplasmic vacuoles at 120 h. Although foci of myocytolysis were present in the CLP + NC group at 24 h, no morphological changes were observed at 120 h.

#### 3.2.2. Ultrastructural Analysis of the Myocardium

Evaluation of the ultrastructure of cardiac muscle cells showed alterations in the CLP group (Figure 3A,B), including lysis, mitochondrial edema, interfibrillar edema, disorientation, and fragmentation of myofibrils. In contrast, in the treated groups [CLP + FC (Figure 3C,D) and CLP + NC (Figure 3E,F)], although the mitochondria and myofibrils were preserved, lipid vacuoles were present.

### 3.3. Gene Expression of mTORC1 Complex Components

Gene expression analysis of mTORC1 complex components showed significantly lower expression (*p* < 0.001) in the untreated septic group (CLP) at 24 h than in the SHAM group (Figure 4A). In contrast, no statistical difference in expression was observed in the CLP + FC group at 24 h. However, mTORC1 expression was significantly lower (*p* < 0.05) in the CLP + NC group than in the CLP group (Figure 4A). A comparison of mTORC1 expression at the 24 h and 120 h time points showed that expression in the CLP group was significantly higher (*p* < 0.05) at 120 h (Figure 4A), but was significantly lower in the treated septic groups (CLP + FC and CLP + NC; *p* < 0.001 and *p* < 0.001, respectively) than in the CLP group at 120 h (Figure 4A).

Raptor expression at 24 h did not differ between the untreated septic mice (CLP) and the treated septic mice (CLP + FC and CLP + NC, Figure 4B). In contrast, compared to the levels at 24 h, Raptor expression had increased significantly (*p* < 0.01) at 120 h in the septic mice (CLP; Figure 4B). The groups of septic-treated mice (CLP + FC and CLP + NC) showed significant reductions in expression (*p* < 0.01 and *p* < 0.001, respectively) compared to the CLP group (Figure 4B).

### 3.4. Gene Expression of mTORC2 Complex Components

Gene expression analysis of mTORC2 components showed that sepsis did not alter the gene expression of this complex, since there was no difference between the untreated septic mice (CLP) and the control mice (SHAM) at 24 or 120 h (Figure 5A). Treatment with curcumin (CLP + FC and CLP + NC) led to significant increases (*p* < 0.05 and *p* < 0.0001, respectively) in expression at 24 h, and when compared with the expression at 120 h, a significant reduction (*p* < 0.05 and *p* < 0.0001, respectively) was noted in the septic-treated groups (CLP + FC and CLP + NC; Figure 5).

In contrast, sepsis altered the expression of Rictor—its expression was higher (*p* < 0.0001) in the septic group (CLP) than in the control group (SHAM) at 24 h but not at 120 h after the induction of sepsis (Figure 5B). The septic mice treated with free curcumin (CLP + FC) had lower Rictor expression at 24 h than the mice in the CLP and CLP + NC groups (*p* < 0.0001 and *p* < 0.0001, respectively, Figure 5B). Expression in the CLP + NC group was lower (*p* < 0.01) than that in the CLP group and higher (*p* < 0.0001) than that in the (CLP + FC) group at 24 h (Figure 5B). When the expression of Rictor mRNA was compared at the two time points (24 and 120 h), the expression of the CLP group was significantly increased (*p* < 0.0001) at 24 h, but not at 120 h, suggesting that sepsis did not affect the expression of this gene at 120 h after sepsis induction (Figure 5B). A similar pattern was observed in the treatment groups (CLP + FC and CLP + NC), with decreased expression (*p* < 0.05 and *p* < 0.01, respectively) at 120 h (Figure 5B).

### 3.5. Gene and Protein Expression of Total mTOR

Analysis of mTOR gene expression revealed that sepsis did not alter its expression at 24 h, and there was no difference between the control (SHAM) and untreated septic (CLP) mice. However, in the treated septic mice (CLP + FC and CLP + NC), mTOR levels were significantly lower (*p* < 0.05, both) than those in the CLP mice (Figure 6A). When we compared mTOR levels at 24 and 120 h after the induction of sepsis, we observed a decrease in mTOR levels in the CLP group (*p* < 0.001; Figure 6A) and increases in the treated septic groups (CLP + FC and CLP + NC) at 120 h (*p* < 0.05, both; Figure 6A).

Analysis of the protein expression revealed no statistical differences among the groups at 24 h (Figure 6B). However, at 120 h, there was a significant increase (*p* < 0.001) in the septic group (CLP) compared to the control group (SHAM; Figure 6C). A significant increase (*p* < 0.001) was also observed in the CLP + NC group compared to the untreated septic group (CLP; Figure 6C).

## 4. Discussion

In this study, we demonstrated, for the first time, the effects of curcumin on components of the mTOR pathway (total mTOR, mTORC1, Raptor, mTORC2, and Rictor) in the heart of an experimental model of sepsis. Along with these mTOR-related changes in curcumin-treated mice, we also observed a reduction in cardiac lesions. These data support our hypothesis that mTOR is a target pathway in sepsis and its regulation may help to preserve cardiomyocytes during sepsis.

The main changes caused by sepsis in the mice at 120 h using the Murine Sepsis Score, along with survival, were body weight changes and temperature variations between the model mice with untreated sepsis (CLP) and the septic mice treated with free curcumin (CLP + FC) or nanocurcumin (CLP + NC). Silva et al. (2017) also observed no statistical differences in the survival of CLP rats treated with curcumin [32]. However, some researchers have shown that curcumin treatment significantly increased survival in septic mice [33,34,35]. Here, we used a less invasive administration route (subcutaneous) for curcumin to reduce animal manipulation during sepsis induction, and our treatment was administered only once, right after the CLP procedure. These factors may have influenced the absorption and bioavailability of curcumin during the evaluation period (120 h). Furthermore, as sepsis is a complex, highly heterogeneous syndrome, a single compound or drug may not be sufficient to control all systemic alterations, even in an experimental model.

Some mechanisms have already been proposed to explain the cytoprotective action of curcumin, including the anti-inflammatory effects of reducing the release of proinflammatory mediators, such as IL-6, IL-1, and TNF- α, and inhibiting NF-κB [36,37,38,39,40,41,42]. Antioxidant action is also another potential contributor, as curcumin has high levels of antioxidant activity and inhibits lipid peroxidation, thus protecting the cell membrane [43,44,45,46]. In agreement with other studies, we observed that curcumin, both in its free form and as nanocurcumin, maintained the integrity of the heart tissue during sepsis [45,47]. Therefore, our hypothesis is that the mTOR pathway may be directly related to the pathogenesis of SIMD, since it is related to cell survival—its modulation by curcumin (FC and NC) could be another important cytoprotective mechanism.

mTOR responds to various stressors, such as tissue hypoxia, inflammatory mediators, and low energy availability, which are involved in cardiac impairment during sepsis [6,21,48,49]. Because mTOR is sensitive to cellular changes, it is necessary to be cautious in both its stimulation and its inhibition, as it can have mixed effects. For example, Pulakat et al. (2017) reported that chronic inhibition of mTOR by rapamycin reduced obesity and cardiac fibrosis in obese mice but increased blood glucose levels. In addition, rapamycin treatment induced cardiac fibrosis in healthy mice when compared to mice with diabetes [50,51].

We investigated how components of the mTOR pathway are affected by the induction of experimental sepsis and the administration of curcumin. We observed that sepsis altered mTORC1 gene expression at both 24 and 120 h. This was expected as it is known that this complex responds to inflammatory mediators, hypoxia, and amino acid availability [12,15,16]. Inhibition of the mTORC1 complex may allow cardiac cells to initiate autophagy, thus reducing cell death [21]. Administration of nanocurcumin reduced mTORC1 gene levels at 24 and 120 h. In contrast, free curcumin only elicited this effect at 120 h. This difference may be due to better cellular uptake compared to free curcumin, which is a hydrophobic compound with more limited absorption.

Suppression of mTORC1 by curcumin may occur via two distinct mechanisms: interruption of the association between Raptor and mTOR and activation of AMPK, a negative regulator of mTORC1 [27,28,52,53,54,55]. Here, we observed a reduction in mTORC1 gene levels in septic mice treated with nanocurcumin at 24 h but did not observe a similar reduction in Raptor at this time point, suggesting that curcumin may be acting on AMPK. However, more robust investigations are necessary to fully understand the mechanism underlying this effect.

In addition to regulating the production of reactive oxygen species and mitochondrial respiration, the mTORC2 complex is also indispensable in the heart, since its absence impairs the differentiation and electrophysiology of cardiomyocytes [56,57,58,59]. Although sepsis did not alter mTORC2 gene expression, the mice treated with both formulations of curcumin had higher levels of this gene than the untreated septic mice at 24 h, but not at 120 h. The reduction in cardiac lesions observed at 120 h may reflect this increased mTORC2 expression in the septic mice treated with curcumin since the activation of mTORC2 after myocardial infarction, induced by coronary artery occlusion, reduced apoptosis of cardiomyocytes [58].

Surprisingly, increased Rictor gene expression observed at 24 h after the induction of sepsis was also observed in the septic mice treated with nanocurcumin but not in the septic mice treated with free curcumin. Rictor is a major protein of the mTORC2 complex and plays crucial roles in cardiac development, cytoskeleton organization, and cell survival [57]. Embryonic Rictor knockdown cells showed reduced translocation of connexin-43 to mitochondria and impaired mitochondrial function [57], indicating that Rictor regulates the translocation of connexin-43 to mitochondria in embryonic cardiomyocytes.

Several hypotheses for SIMD have been proposed, including oxidative stress, tissue hypoxia, and loss of cardiac structural proteins. However, we believe that no single factor or pathway can explain the sepsis-induced cardiac dysfunction in SIMD, but rather propose that SIMD is a result of several simultaneous changes—the data presented here add another piece to this puzzle.

## 5. Conclusions

Our results indicate that treatment with free curcumin and nanocurcumin reduced the cardiac lesions induced by severe sepsis in the model mice and helped to maintain the structural integrity of mitochondria and cardiac myofibrils. Free curcumin and nanocurcumin altered components of the mTOR pathway, specifically mTORC1, Raptor, mTORC2, and Rictor, in the heart of the septic mice. Our findings suggest that the mTOR pathway may be directly related to sepsis-induced cardiac injury and proper modulation of mTOR may be a protective mechanism.

## Figures and Tables

**Figure 1 pharmaceutics-14-02277-f001:**
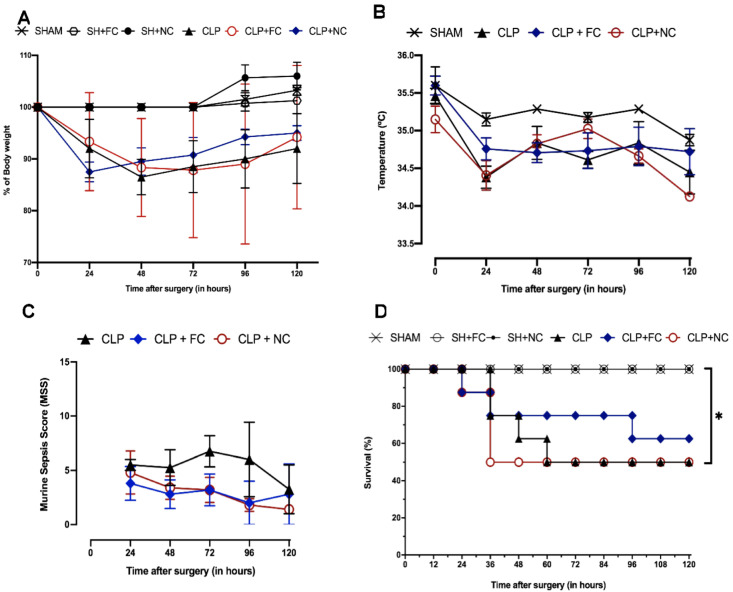
Characterization of murine sepsis by cecum ligation and puncture (CLP). In (**A**), find the body weight in percentage over the evaluation period. In (**B**), the temperature is observed in degrees Celsius (°C). The graph in (**C**) shows the values referring to the murine sepsis score evaluated daily. In (**D**), the data show the percentage of survival of the animals during 120 h. Animals were followed daily for 120 h after sepsis induction (n = 8 per group). *p* values * *p* < 0.05.

**Figure 2 pharmaceutics-14-02277-f002:**
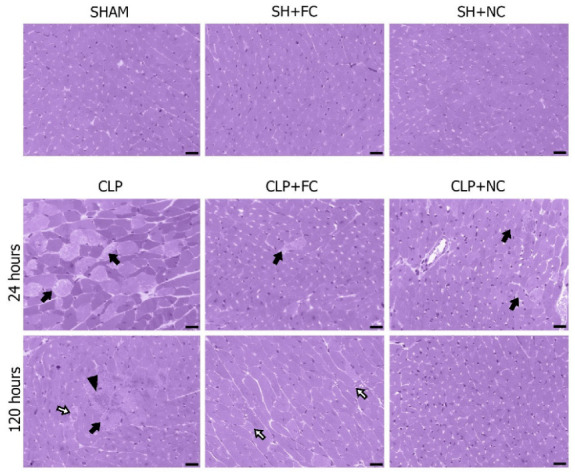
Histopathology of myocardial tissue from mice subjected to cecal ligation and puncture (CLP) sepsis. The myocardium of the sham-operated animals without SHAM treatment and treated with free curcumin (SH + FC) or nanocurcumin (SH + NC) after 120 h did not show any histopathological changes. Analysis of the myocardium in the septic group shows the presence of accentuated and diffuse myocytolysis (black arrow) at 24 h, and at 120 h, foci of cytoplasmic vacuoles (white arrow), and multifocal myocytolysis (black arrow) associated with bands of contracture (arrowhead). In the septic group treated with free curcumin (CLP + FC), at 24 h the presence of mild focal myocytolysis is noted (black arrow); at 120 h, cytoplasmic vacuole foci are observed (white arrow). In the septic group treated with nanocurcumin (CLP + NC), at 24 h it is possible to observe mild focal myocytolysis (black arrow); at 120 h it is possible to notice that the cardiac tissue is preserved. Photomicrographs were obtained using a 40× objective (50 µm).

**Figure 3 pharmaceutics-14-02277-f003:**
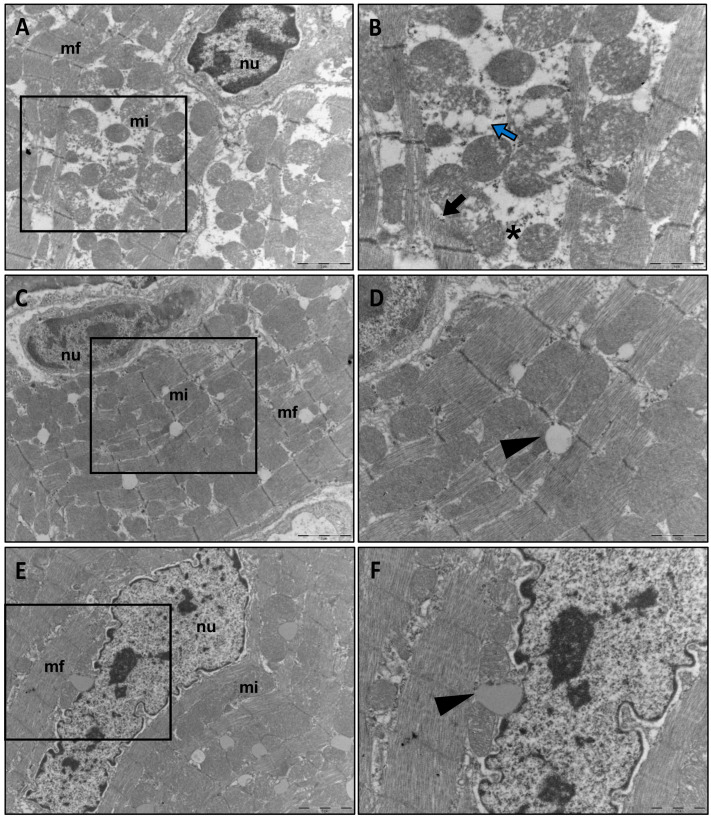
Myocardial ultrastructural changes in CLP-induced sepsis. Assessment of the ventricular myocardium in the untreated CLP septic group after 120 h (**A**,**B**) reveals edema and mitochondrial lysis (blue arrow), in addition to interfibrillar edema (asterisk), myofibrillar rupture, and disorientation (black arrow in (**B**)). The treated groups CLP + FC (**C**,**D**) and CLP + NC (**E**,**F**) showed preserved mitochondria and myofibrils, with some lipid vacuoles (arrowhead). In (**A**,**C**,**E**) the bars indicate 2 µm, and in (**B**,**D**,**F**) 1 µm.

**Figure 4 pharmaceutics-14-02277-f004:**
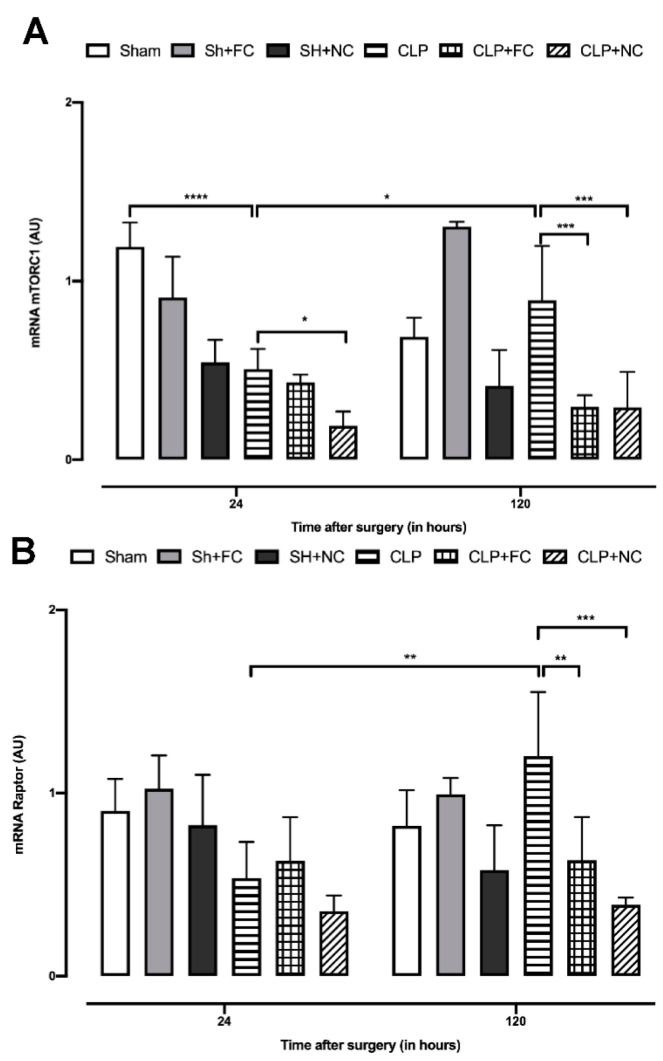
Real-time PCR gene expression profile analysis of mTOCR1 and Raptor in the heart of curcumin-treated and non-treated septic animals. (**A**) Assessment of mTORC1 mRNA levels shows that the septic group (CLP) had a significant reduction (*p* < 0.001) compared to the control (SHAM) at 24 h and an increase at 120 h (*p* < 0.05). The group of septic animals when treated with nanocurcumin (CLP + NC) showed a reduction in mTORC1 when compared to (CLP) at 24 and 120 h (*p* < 0.05 and *p* < 0.001, respectively). At 24 h, there was no difference in the septic group treated with free curcumin, but at 120 h there is a significant reduction (*p* < 0.001) compared to (CLP). (**B**) Raptor analysis shows that there was no difference between either group 24 h after sepsis induction. When comparing the levels of Raptor at 24 h with 120 h, it is noted that there was a significant increase in the (CLP) group (*p* < 0.01). At 120 h, there is a reduction in Raptor in the treated septic groups (CLP + FC and CLP + NC) compared with the untreated septic group (CLP) (*p* < 0.01 and *p* < 0.001, respectively). Results are expressed in arbitrary units (AU). *p* values * *p* < 0.05; ** *p* < 0.01; *** *p* < 0.001; **** *p* < 0.0001.

**Figure 5 pharmaceutics-14-02277-f005:**
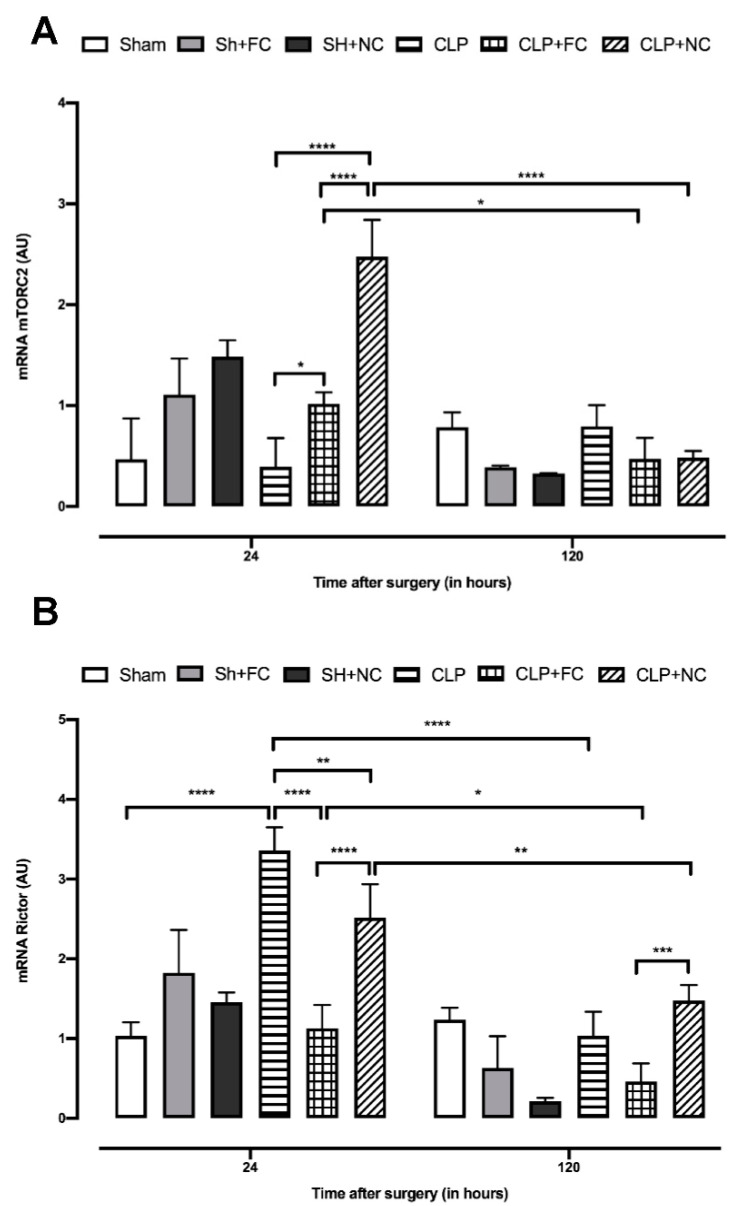
Real-time PCR gene expression profile analysis of mTOCR2 and Rictor in the heart of curcumin-treated and untreated septic animals. (**A**) The assessment of mTORC2 mRNA levels shows that the treated septic groups (CLP + FC and CLP + NC) had a significant increase (*p* < 0.05 and *p* < 0.0001, respectively) compared to the untreated group (CLP) at 24 h. Analysis at 120 h showed that there was no significant change in the septic group (CLP), but there is a reduction in the treated septic groups (CLP + FC and CLP + NC) at 120 h when compared to the expression at 24 h (*p* < 0.05 and *p* < 0.0001, respectively). (**B**) Rictor’s analysis shows that sepsis (CLP) induced a significant increase (*p* < 0.0001) in the expression of this gene at 24 h when compared to the control (SHAM). In the treated septic groups (CLP + FC and CLP + NC), there is a decreased expression (*p* < 0.0001 and *p* < 0.0001, respectively) of Rictor when compared to the untreated septic group (CLP) at 24 h. At 120 h, there was a reduction in the expression of this gene in the septic group (CLP) when compared to 24 h (*p* < 0.0001). Likewise, Rictor expression in the treated septic groups (CLP + FC and CLP + NC) decreased (*p* < 0.05 and *p* < 0.01, respectively) compared to 24 h. Furthermore, the septic group treated with free curcumin (CLP + FC) showed a significant decrease (*p* < 0.001) compared to the group treated with nanocurcumin (CLP + NC). Results are expressed in arbitrary units (AU). *p* values * *p* < 0.05; ** *p* < 0.01; *** *p* < 0.001; **** *p* < 0.0001.

**Figure 6 pharmaceutics-14-02277-f006:**
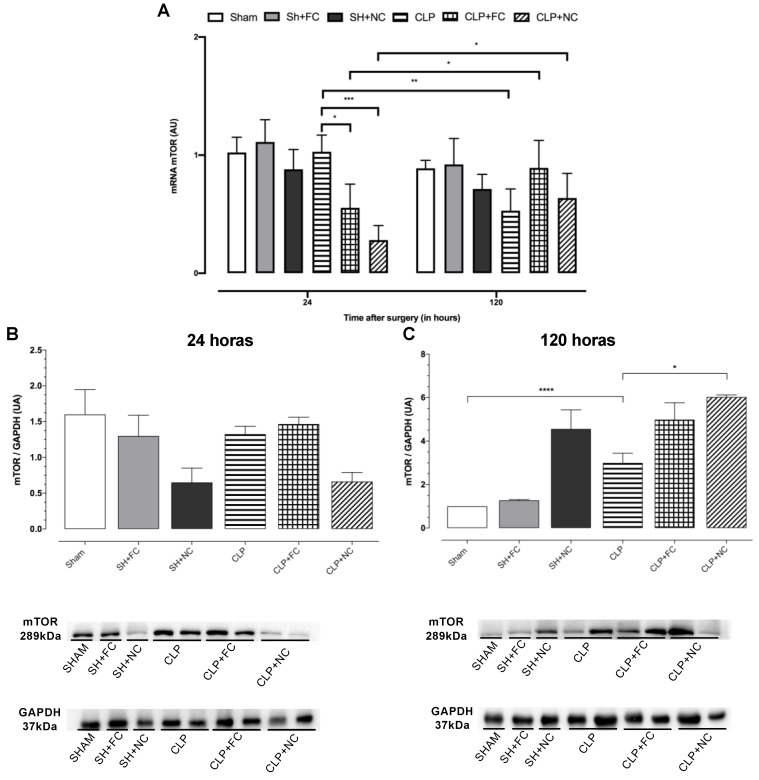
Analysis of gene expression profile by real-time PCR and protein by Western blotting of mTOR in the heart. (**A**) At 24 h there is no change in mTOR gene expression in the (CLP) group when compared to the control (SHAM); however, in the treated septic groups (CLP + FC and CLP + NC) (*p* < 0.05 and *p* < 0.001, respectively), there was a significant reduction compared to (CLP). After 120 h of sepsis induction, there was a significant reduction in mTOR in the (CLP) group (*p* < 0.01) compared to at 24 h; on the other hand, the treated groups (CLP + FC and CLP + NC) (*p* < 0.05 and *p* < 0.05, respectively) increased. (**B**) No significant differences were found in the mTOR protein levels in the myocardium of the animals at 24 h. (**C**) At 120 h, there was a significant increase in the CLP group (*p* < 0.0001) compared to the control (SHAM) in the CLP + NC group (*p* < 0.05) in relation to the untreated septic group (CLP). At the bottom of (**B**,**C**), the autoradiograph resulting from western blot analysis of representative protein levels for mTOR and GAPDH of mouse hearts, subjected to sham operation (SHAM, SH + FC, and SH + NC) or sepsis induction (CLP, CLP + FC, and CLP + NC) 24 and 120 h after surgery. Results are expressed in arbitrary units (AU). *p* values * *p* < 0.05; ** *p* < 0.01; *** *p* < 0.001; **** *p* < 0.0001.

**Table 1 pharmaceutics-14-02277-t001:** Description of experimental groups.

Experimental Groups	
SHAM	group of sham-operated animals;
SH + FC	group of sham-operated animals treated with free Curcumin;
SH + NC	group of sham-operated animals treated with nanocurcumin;
CLP	group of animals submitted to a severe septic stimulus;
CLP + FC	group of animals submitted to severe septic stimulus treated with free Curcumin;
CLP + NC	group of animals submitted to severe septic stimulus treated with nanocurcumin;

**Table 2 pharmaceutics-14-02277-t002:** PCR primer sequences of mTOR pathway and GAPDH gene in mouse for quantitative polymerase chain reaction (qPCR).

Gene	Forward	Reverse
GAPDH	CTTTGTCAAGCTCATTTCCTGG	TCTTGCTCAGTGTCCTTGC
mTORC1	TCGATGAATGTGGGATTGTGG	TGCCTTCGCTGGAGAATATC
mTORC2	ATCTCCGTGTTTATGCTGTCC	CACCGTTTCTCCATTGAGAAC
Raptor	GCAGAGCTGGAGAATGAAGG	GTCGAGGCTCTGCTTGTACC
Rictor	ATGGAAATAAGGCGAGGTCTG	AAAGCCTCCAACTGTCCTG

## Data Availability

Not applicable.

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
