# Peer review of "Modulation of the mTOR Pathway by Curcumin in the Heart of Septic Mice"

_pharmaceutics, 2022, doi:10.3390/pharmaceutics14112277_

Round 1

Reviewer 1 Report

Authors reported the usage of free curcumin (CL) and nanocurcumin (NC) to evaluate the action of curcumin on the mTOR pathway in the heart of septic mice. This article has done some work with some data. However, to be considered for publication, authors should make major revisions of this manuscript:

1. The introduction needs to add examples of related CL and NC to evaluate mTOR pathway, and show the advantages and innovation of this article by comparison.

2. The article lacks of materials information. And detailed instrument information should be given.

3.  Some figures are not clear.

4.  English and format needs to be improved because of many errors.

5.  Does the authors test the long time stability of NC. 

Author Response

Reviewer #1:

Report

Authors reported the usage of free curcumin (CL) and nanocurcumin (NC) to evaluate the action of curcumin on the mTOR pathway in the heart of septic mice. This article has done some work with some data. However, to be considered for publication, authors should make major revisions of this manuscript:

  1. The introduction needs to add examples of related CL and NC to evaluate mTOR pathway, and show the advantages and innovation of this article by comparison.

Thank you for your insightful comment. Based on your observations, we have made careful changes to the manuscript introduction and a paragraph was added to the text (changes were highlighted in yellow).

  1. The article lacks of materials information. And detailed instrument information should be given.

We agree with the reviewer, the methodology was revised and the missing points were added and are highlighted in yellow in the text.

  1. Some figures are not clear.

Thank you for your insightful comment. All original western blot images and histopathology figures were sent separately by e-mail to the Pharmaceutics Editorial Office.

  1. English and format needs to be improved because of many errors.

Thanks for your comment and recommendations, the text has been revised by the Edtage website and has undergone new revision.

  1. Does the authors test the long time stability of NC.

We agree with the reviewer's recommendation and this information was inserted into the text. Our study followed the protocol to obtain nanocurcumin as published by BHAWANA et al. (2011) (doi: 10.1021/jf104402t), which demonstrated the stability of this formulation. BHAWANA et al. (2011) demonstrated that freeze-dried nanocurcumin is stable at room temperature for 6 months without any decomposition or aggregation.

Reviewer 2 Report

1. What was the basis for the selection of doses of Free curcumin and nano curcumin (NC), both at a dose of 12.5 mg?

2. The water solubility of curcumin is almost nil. How the authors diluted in sterile 0.9% NaCl saline solution (100 uL volume/animal)?

3. Why has the route oral not been opted and what is the rationale for injecting subcutaneously?

4. What was the site of the SC injection?

5. Which ANOVA was used and why Tukey’s test was preferred over others?

6. Use a better abbreviation for free curcumin.

7. Better H & E images with better resolution should be provided/

8. The scale bar in the images is missing.

9. Sh + CL can be better presented as SH.

10. Cardiac function data is missing.

11. Cardio-specific enzyme markers are missing.

Author Response

Reviewer 2

  1. What was the basis for the selection of doses of Free curcumin and nano curcumin (NC), both at a dose of 12.5 mg?

The concentration of curcumin used in our study was based on a vast bibliographic survey that culminated in the publication of the review "Curcumin as a Potential Treatment for COVID-19" (doi.org/10.3389/fphar.2021.675287) and in pilot studies carried out by our research group. In these pilot experiments, curcumin was administered by gavage immediately before, and every 12 hours after experimental sepsis induction using a CLP model for 5 days. Our results demonstrated that animals treated with curcumin improved survival rate and murine sepsis score when the cumulative concentration reached 12.5 mg/kg. Therefore, we opted for this concentration. In addition, the subcutaneous route of curcumin administration was chosen because it is less aggressive to the animal and generates less trauma compared to the various administrations of curcumin by gavage.

  1. The water solubility of curcumin is almost nil. How the authors diluted in sterile 0.9% NaCl saline solution (100 uL volume/animal)?

Taking into account and knowing that the solubility of curcumin in water is lower, during the applications, the curcumin solutions (free and nanocurcumin) were homogenized with the syringe used for administration. It is worth mentioning that nanocurcumin has a higher degree of solubility in water, as previously described by BHAWANA et al. (2011) (doi: 10.1021/jf104402t) and, additionally, other studies have also demonstrated greater solubility and stability of nanocurcumin, according to Gopal et al. (2016) (doi: https://doi.org/10.1016/j.foodchem.2016.05.140) and Shariati et al., 2019 (doi: 10.2147/IDR.S213200).

  1. Why has the route oral not been opted and what is the rationale for injecting subcutaneously?

In this study, the subcutaneous route was chosen with the main objective of reducing the handling and suffering of animals with severe sepsis. Furthermore, for oral use, multiple administrations would be required during the study period (120 hours or 5 days) as curcumin is not well absorbed from the intestinal tract (Kurita T, Makino Y. Novel curcumin oral delivery systems. Anticancer Res.2013 Jul;33(7):2807-21. PMID: 23780965.). Thus, the subcutaneous route was used to provide more effective and sustained tissue concentrations (Prasad et al., 2014 [doi: 10.4143/crt.2014.46.1.2]).

  1. What was the site of the SC injection?

Thanks for your comment and recommendations. The site for injection was on the dorsum (over the shoulders) into the loose skin over the neck.

  1. Which ANOVA was used and why Tukey’s test was preferred over others?

We agree with the reviewer's recommendation and this information was inserted into the text. In our study, initially, the results were analyzed with Student's t-test to assess whether the distribution was normal (parametric data). Statistical analysis was evaluated by the one-way ANOVA method with the Bonferroni post-test. Bonferroni correction was used to reduce the chances of getting false-positive results (Type I errors) when multiple pair tests are performed on a single dataset.

  1. Use a better abbreviation for free curcumin.

Thanks for your observation, this abbreviation has been improved in the text.

  1. Better H & E images with better resolution should be provided/

The original images were forwarded to the magazine, thanks for your suggestion.

  1. The scale bar in the images is missing.

Thanks for your observation, the bars have been provided.

  1. Sh+CL can be better presented as SH.

Thanks for your comment, the abbreviation SH+CL refers to the sham-operated group treated with free curcumin, so we don't just use "SH".

  1. Cardiac function data is missing.

We agree with the reviewer's recommendation, but unfortunately, during this study, it was not possible to evaluate cardiac function due to the unavailability of suitable equipment for mice. In addition, due to the pandemic period, the availability of animals for experimentation was reduced, making it difficult to carry out the experiments.

  1. Cardio-specific enzyme markers are missing.

Thank you for your insightful comment. At the beginning of the project, the measurement of cardiac biomarkers was planned, however, due to the pandemic period, the reduced availability of animals for experimentation, and the logistical and financial difficulties in acquiring commercial kits, it was impossible to carry out the measurements.

Round 2

Reviewer 2 Report

Can be accepted.